



# JSP
## Job Search Platform

**Autors**: Mateusz Molenda · Bohdan Kyryliuk · Franciszek Lepka · Sergiy Vergun

**Supervisor:** Rafał Palak PhD, Wrocław University of Technology

### Abstract

This paper presents the development of a scalable recruitment platform designed to bridge the gap between recruiters and job seekers. Using a microservices architecture, the platform ensures high availability and scalability to adapt to a growing user base without performance degradation. Key features include custom quizzing mechanisms for pre-evaluation of the candidates, integrated chat services for streamlined communication, and analytical tools providing market insights and user behavior trends. The platform also incorporates a recommendation algorithm to enhance job matching for candidates. The project aims to improve the recruitment process by saving time for recruiters and providing valuable insights to job seekers.

## 1 INTRODUCTION

In today's fast-paced job market, the recruitment process can be time-consuming and inefficient for both recruiters and job seekers. Recruiters often face challenges in sifting through numerous applications to find suitable candidates, while job seekers struggle to find positions that match their skills and preferences. This project aims to develop a scalable recruitment platform that addresses these challenges by leveraging modern technologies and design principles.

The primary objective is to create a platform that adapts seamlessly to a growing number of users without compromising performance. By employing a microservices architecture designed with scalability in mind, the platform ensures high availability and reliability. The target solution chosen for that purpose was Kubernetes - as it is open source, with verified and used in production by many companies across the world. The microservices architecture allowed for more flexible development, by allowing to split work on separate microservices to individual team members. Additionally it enabled us to utilize different programming languages, to better accommodate to given microservice's task. For example main application logic is written in Kotlin using Spring Boot framework, the analytics micro-service is written in Python programming language, which allows for very easy data manipulation and processing. Lastly for chat service we decided to use Go programming language due to its low learning curve and simple but robust design.

The project focuses on closing the gap between recruiters and job seekers by providing tools that enhance the recruitment process, including custom quizzing mechanisms, integrated chat services, and analytical insights.

The expected business and technical benefits include time savings for recruiters through pre-evaluation tools, improved communication channels, and valuable market insights for both parties. Job seekers benefit from a recommendation algorithm that personalizes job suggestions based on offers they already interacted with.

## 2 RELATED WORK

Existing recruitment platforms such as LinkedIn, Indeed, and Glassdoor offer various features for job searching and candidate sourcing. However, they don't include a verification/testing system, which is often being outsourced by companies for their specific job offers. Only LinkedIn came close with general skills assessments for a candidates, so that they could show on their profile that they've finished the test for given skill, eg. advanced Java, but they've resigned from that solution, to instead tag a skill to specific job a user's had in the past. Another missing part is market analytics, which is often not present on the website or hidden behind some requirements.

The choice of a microservices architecture addresses the limitations of monolithic systems in handling scalability and high availability. By decomposing the platform into independent services, we can ensure that each component is optimized and can scale independently based on demand.

## 2.1 Proposed solution

For our project we decided to use microservices infrastructure - which allows for easier scaling in case of rapid increase in the user base. With that covered, we wanted to implement the following features: custom quizzing mechanism, real-time analytics, and integrated communication tools.

Our project distinguishes itself by integrating a custom quizzing mechanism that allows recruiters to pre-evaluate candidates using their own assessments. This feature streamlines the candidate selection process and reduces the time spent on initial screenings. In addition, the platform offers integrated chat services for direct communication and advanced analytics to provide insights into market trends and user behavior.

### 2.1.1 Core Features

As for the main features we wanted to implement:

- robust authentication/authorization system to distinguish between applicants/recruiters

- custom quizzes mechanism

- analytics graphs about current job market state

- recommendation algorithm

- integrated chatting system

- CV generation tool

### 2.1.2 Technical Approach

Scalable Microservices Architecture

The adoption of a microservices architecture ensured that the platform could handle a growing number of users without performance degradation. Each service can scale independently of others, thanks to the implementation of load balancing via Kubernetes Service objects, the platform achieves high availability, minimizing downtime and ensuring a reliable user experience. As the base requirement to be using the Kuberentes technology, we established that we'll use Docker for deploying our microservices. After requests go through the API Gateway they are redirected towards their specific microservices (each can have a number of running stateless containers behind a load balancer. As the only stateful part being the database we scaled it in a simpler way: by splitting it to master and read-only replicas, thanks to which we can scale our database horizontally when it comes to read operations (visiting website).

Custom Quizzes Mechanism

Recruiters can create and reuse custom quizzes to with varying number of answers per question, additionally defining whether the question is single choice or multiple choice. When recruiter creates a job offer he can specify the quiz, which the applicants should take for this specific job offer. Then a recruiter can overview the top N (N - natural number $0 <= N <=$ number of applicants) applicants and sort them by their score or time it took to finish the quiz. This feature allows for efficient filtering of applicants based on skills and competencies specific to the job requirements. It can serve as great pre-verification mechanism to decrease the number of potential candidates base on the preliminary domain-knowledge test.

Integrated Chat Service

To allow for easier communication, we implemented the chat service for the recruiter to be able to contact with interesting candidates easier. The chat is available to all users, but only the recruiters can initiate a chat with an applicant, in this way they control who they communicate with, and don't receive unimportant spam. An integrated chat service facilitates direct communication between recruiters and selected candidates. This feature improves engagement and speeds up the recruitment process by allowing immediate feedback and scheduling of interviews.

### 2.1.3 Innovations

One of the few of the non-standard approaches we took to enhance the quality of our application: database master-slaves model: where we can create more read-only database instances, which can be beneficial when the number of read requests grows much more than write operations, but also in case where we could meet the limit on disk I/O speed on a singular machine.

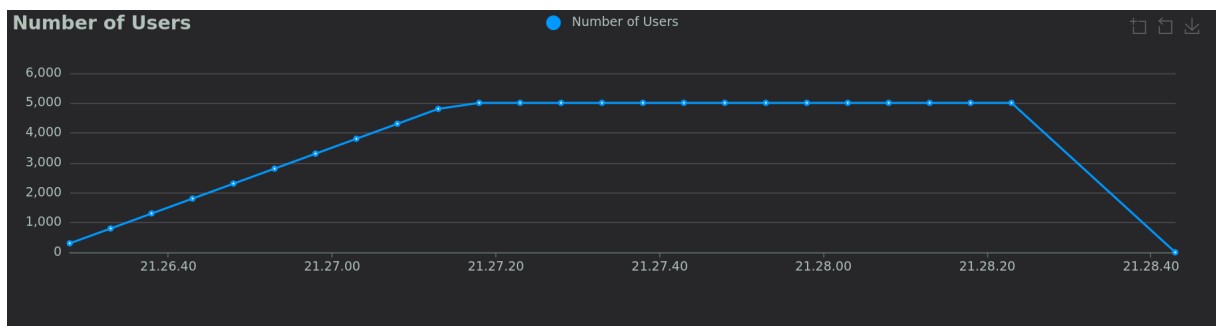

Figure 1: Number of active users

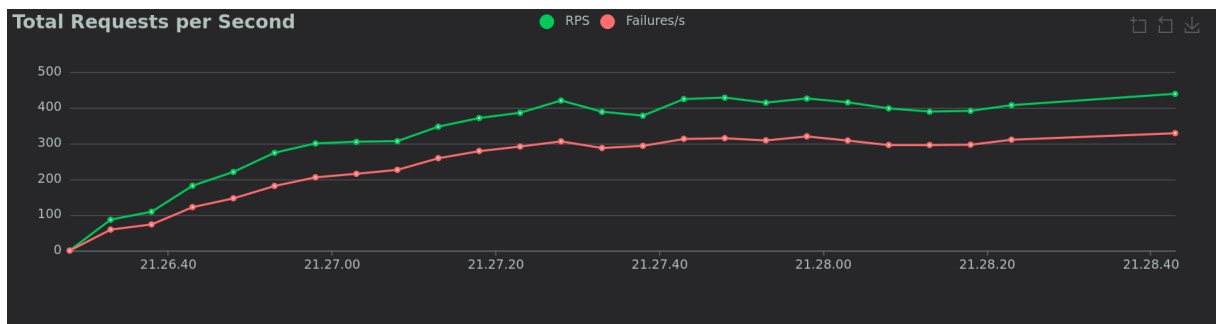

Figure 2: Number of requests per second

### 2.1.4 Benefits and Impacts

Our most outstanding features are custom quizzing and resume generator which help to increase the user experience by providing out-of-the-box solutions for some of the problems visible in other job searching websites.

### 2.1.5 Challenges and Mitigation

During this project we've had 2 big challenges: First one was at the beginning of our working during summer holidays (July - September), where our original estimates failed. Originally it was time to create at least 50% of the project, starting from base documentation, database schema, design mockups and proof of concepts for the technologies we wanted to use. Due to the fact that all our team members were working full-time, as well as having various holiday plans, the work done was vastly overestimated, and looking back it can be said to be about 15% of the actual code. All things considered it was very good that we've started those 2 months earlier as in this time we started working on new technologies: Kotlin, GraphQL, k8s, horizontal database scaling, which took a lot of time to properly research and implement the base working examples. Second problem was unplanned bad health condition and later sick leave of our core Frontend Developer, due to which we've found ourselves understaffed for all the stuff we've wanted to implement. That resulted in potential worse quality and increased number of bugs in various parts of our system.

## 3   RESULTS

The project successfully developed a MVP (Minimum Viable Product) of scalable recruitment platform
To verify that our solution is performant, we performed simple stress tests:
In this case, to simplify the deployment to the cloud, we used a single `e2-medium` instance on Google Cloud with 2 cores (4 threads) and 4 GiB of RAM, and ran our application using `docker compose`.
The tests were performed using *Locust tool* [1], with 3 worker nodes, all running from one machine. The peak number of concurrent users was set to 5000 (Figure 1), with 100 users added per second. Additionally, each user had a defined random timeout of 5–10 seconds to simulate more realistic user behavior.
Most requests were particularly designed to be unauthorized, resulting in a notable number of failures (Figure 2). Despite significant delays, the server managed to retain decent latency at least for the 50th percentile (Figure 3). Resource usage remained relatively low even during peak requests (Figure 4).

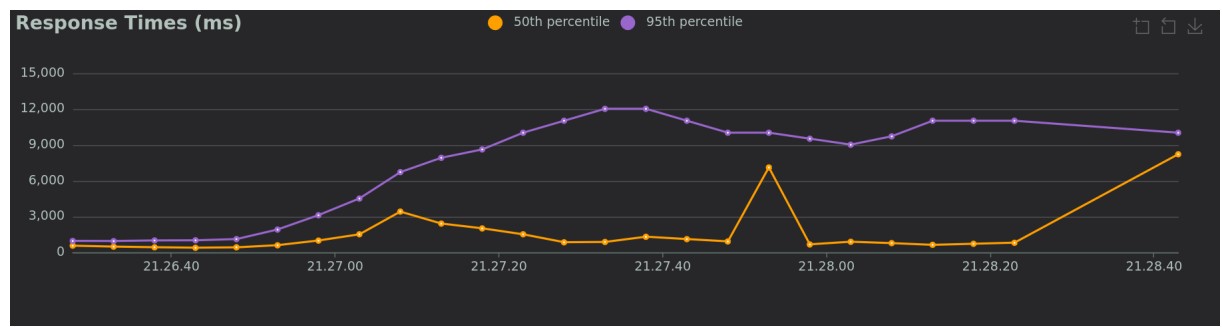

Figure 3: 50th and 90th percentiles latency in ms

```
32d935d529d8   job_market_user_service      47.00%   819.7MiB / 3.833GiB    20.88%   13MB / 15MB       69.1MB / 2.49MB    230
4c32b1284add   job_market_notification       0.13%   179.2MiB / 3.833GiB     4.57%   28.6kB / 29.4kB   31.5MB / 1.93MB     40
b5ccbcc3df69   job_market_job_service       24.42%   424.8MiB / 3.833GiB    10.82%   149MB / 210MB     74.2MB / 1.85MB     67
d9888ae1abf0   job_market_gateway           46.64%   786.5MiB / 3.833GiB    20.04%   247MB / 271MB     12.8MB / 1.97MB     42
306f882e1194   rabbitmq                      0.18%   109.5MiB / 3.833GiB     2.79%   32.7kB / 26.1kB   22.3MB / 606kB      26
b302271a6c1a   job_market_database           7.55%   71.84MiB / 3.833GiB     1.83%   18.1MB / 130MB    19.1MB / 59.9MB     27
```

Figure 4: Resource usage during peak 400 requests/second

### 3.1   Analytical Insights

The platform provides analytical tools that offer insights into market trends, user activity and behaviors on our site. Recruiters and job seekers can access data on companies, users, locations and jobs according to specific parameters such as salary or required experience. Data is presented in a form of interactive charts of many types in site's analytics section. Charts are displayed with help of Chart.js library based on JSON files generated by analytics server. JSON files are generated once a day on a fixed time and saved in a database. Such guarantees up-to-date information for the users and ensures that analytics server is not under heavy stress at all times as it would be if we decided to reevaluate queries for chart data generation every time a user enters analytics section.

### 3.2   Recommendation Algorithm

A recommendation algorithm [2] that is based on content-based filtering [3] method personalizes job suggestions for users by considering job offers they interacted with and comparing them with other offers based on offer's parameters. Actions considered as an interaction are applying, following and displaying a job offer. Job offers are vectorized according to parameters and then compared by cosine similarity method. Recommendations improve the job search experience by highlighting relevant opportunities and increasing the likelihood of successful matches.

## 4   TECHNOLOGY STACK

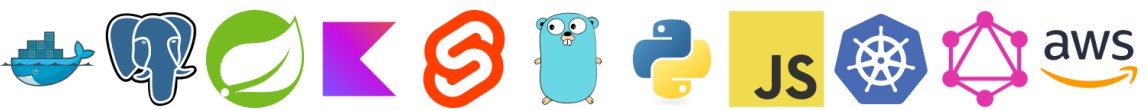

## 5   CONCLUSION

The project achieved its minimal goal of creating a scalable recruitment platform that bridges the gap between recruiters and applicants. We successfully implemented custom quizzes - which is main distinguishing feature of our service, by allowing the recruiters for easier candidates filtering. Analytics provided by our website are great additional feature which makes the service more competitive in comparison to other existing solutions. Additionally, the built-in chat aims to simplify the UX (User Experience) and allow for smoother inter-human interactions.

### 5.1   Lessons learned

Thanks to the broad amount of technologies and solutions we gained valuable insights regarding some of the technologies we used, they are listed here:

### 5.1.1   Authentication

Initially, we selected Keycloak as the authentication solution for our Spring Boot microservices architecture due to its comprehensive feature set, support for industry-standard protocols (e.g., OpenID Connect, OAuth2, SAML), and out-of-the-box functionality. However, after careful evaluation during the integration phase, we identified several challenges that led us to pivot to a custom implementation using JWT [4] (JSON Web Tokens) with Spring Security.
Advantages of the Custom JWT Implementation

- Self-Contained Authentication: JWT tokens are stateless and self-contained, making them ideal for a distributed microservices environment. Our api gateway can independently validate tokens without relying on a central server.

- Reduced External Dependencies: By eliminating the need for an external authentication server, the custom implementation simplifies the deployment process and improves resilience by reducing points of failure.

- Enhanced Flexibility: The JWT-based approach allows seamless integration with our existing infrastructure, including support for custom claims, roles, and dynamic token lifecycles.

### 5.1.2   GraphQL

Originally we were very enthusiastic towards using GraphQL as an alternative to traditional REST API, but the more we worked with it, the more corner cases and issues we saw. The major problem we've had is with authentication, as our website should provide different type of content to non-authenticated and authenticated users, second of which also vary by their role. When researching similar issues, it turns out that such authentication problems are not the only ones [5] when using GraphQL. Even though GraphQL makes it easier for frontend development, and limit the network usage by specifying only the needed fields in each request, it comes at a heavy price of troubles with authentication especially more fine-grained, as well as rate limiting the requests, which can be easily used for DoS (Denial of Service) attacks. Knowing that, we wouldn't have used GraphQL but instead stay with simpler and more reliable REST API.

### 5.1.3   Chart generation

Another important decision was a transition from generating static JPG files using the Plotly library in Python to generating JSON data in Python and rendering charts on the frontend with Chart.js. Such choice was motivated by several factors. First, potential stress on the server was reduced. By sending lightweight JSON data instead of serving image files, the server workload is minimized, allowing for faster responses and improved performance under high user demand. Additionally, Chart.js offers enhanced visual customization options and interactivity compared to static images, leading to a more engaging and dynamic user experience. This approach aligns better with modern web development practices, ensuring a responsive and visually appealing interface.

## 6   FUTURE DIRECTIONS

Future development of the platform could include:

- **Enhanced AI Features**: Incorporating advanced machine learning algorithms for more accurate matching and predictive analytics.

- **Mobile Application**: Developing a native mobile version of the application to increase accessibility and user engagement.

- **Third-Party Integrations**: Integrating with other professional networking tools like Linkedin or Slack.

- **Global Expansion**: Adapting the platform to support multiple languages and regional preferences to cater to a global audience.

## 7   ACKNOWLEDGEMENTS

The authors are grateful to our supervisor, Rafał Palak, PhD, for his invaluable feedback and patience we could always count on, which helped steer the project in the right direction despite many difficulties, as well as to Marcin Maleszka, PhD, and Marcin Jodłowiec, PhD, for their professional hints and advice during the development process.

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
