# OpenReview forum: "Job Search Platform"
_pwr.edu.pl/Wrocław_University_of_Science_and_Technology/2024/ZPI_Day — Wrocław University of Science and Technology 2024 ZPI Day Submission_

### Official Review · Reviewer_eFu1 · 2024-12-05
**Innovative Job Search Platform**

**Confidence:** 5
**Significance Of Results:** 5
**Overall Quality:** 5

**Compliance With Template:**

5: Very High Quality – The article contains all the required sections, which are written in a very detailed, clear, and error-free manner. The structure is professional and meets expectations, and the content adheres to the highest substantive and formal standards.

**Description Of Results:**

5: Very High Quality – The results are described in detail, clearly and comprehensively, supported by thorough evaluation, analysis, and convincing usage examples. The description meets the highest substantive standards.

**Feedback On Consistency:**

The project and its description are coherent; the authors, at the outset of the project, identified gaps in existing solutions and subsequently presented their own approach, effectively addressing each of the previously identified issues. The structure of the article is logical and consistent, facilitating understanding of the presented concepts and their implementation.

**Potential For Development:**

In the section "5.1.1 Authentication," the authors attempted to present their own implementation of JWT as superior to Keycloak. However, their arguments merely highlight the advantages of JWT, which are also utilized by Keycloak. The lack of compelling reasons for the change suggests that this decision was made without fully considering its consequences. The plans for "Global Expansion" are akin to an ambitious journey into uncharted waters – but before embarking, it would be wise to thoroughly analyze the requirements that could help identify complex challenges, such as delivering multimedia content with minimal latency across the globe, where content delivery networks (CDNs) would be invaluable.

**Project Nature Evaluation:**

The authors demonstrated an engineering approach at every stage of project execution, as well as a proficient understanding of best practices, the latest technologies, and tools. Their ability to identify and effectively resolve encountered issues reflects a high level of maturity and a systematic approach to complex challenges. Furthermore, the testing of the developed application confirmed their advanced engineering competencies, serving as evidence of their professionalism and commitment to the project.

The only issue that can be pointed out in the project is the use of a Custom JWT Implementation instead of Keycloak. The authors' reasoning suggests that this decision was either not fully thought through or was based on incorrect assumptions. The lack of a detailed description of the solution prevents a definitive conclusion, but it can be assumed that the Custom JWT Implementation is not as comprehensive as Keycloak would have been. This particularly pertains to compliance with industry standards such as ISO 27001, ISO 27017, and ISO 27018.

**Technical Language Precision:**

5: Very High Quality – The language is entirely appropriate for a technical report. All terms are used correctly and precisely, and the style is professional, clear, and coherent, without any errors or ambiguities.

---

### Official Review · Reviewer_369Y · 2024-12-08
**Solid base, with huge potential to grow.**

**Confidence:** 5
**Significance Of Results:** 4
**Overall Quality:** 4

**Compliance With Template:**

4: High Quality – The article contains all the required sections, which are well-written and substantively correct, although minor errors or shortcomings may be present. The overall structure is clear and coherent.

**Description Of Results:**

5: Very High Quality – The results are described in detail, clearly and comprehensively, supported by thorough evaluation, analysis, and convincing usage examples. The description meets the highest substantive standards.

**Feedback On Consistency:**

The content of the submission seems consistent: the authors decided to develop a system to support the recruiters' work, precisely limiting the scope of the project, without any exaggeration. In the end, the scope of the developed solution is consistent with the starting goal.
One crucial inconsistency: at the top of the paper there is a logo with "Jobistry" text, the paper title is "JSP Job Search Platform". The logo is great but never mentioned in the text. Jobistry is also more safe, as "JSP" is the trademark of Oracle.

**Potential For Development:**

This project has huge potential for development. As authors can extend it in many different directions. The initial decision about using microservices architecture can make adding new features easier.
One direction that should be considered very seriously is the security of the application, as it processes personal data. Authors should take GDPR and compliance with these regulations into account.

**Project Nature Evaluation:**

Yes. The project can be considered engineering work. The authors proved their ability to solve practical and technical problems. They learned new technologies and used them to provide the final solution.
Looking at Figure 4., I can infer that all services job_market* use a single database job_market_database. As authors declared the usage of microservices, they should create independent databases. This database and rabbitmq can become a single point of failure for the system. I would suggest splitting data among databases related to only one microservice and using database and queue services offered by the cloud provider to achieve weaker coupling and higher reliability.
Also, the scalability research shows only one instance of Docker host hence the ability to scale microservices is not used at all. Maybe use of two nodes Docker swarm cluster would provide better results.
The above statements should be considered as a voice in the discussion on the project, possible due to the detailed content of the submission.

**Technical Language Precision:**

4: High Quality – The language is appropriate for a technical report. Terminology is used correctly, and statements are precise, with only minor shortcomings that do not affect the overall clarity.

---

### Official Review · Reviewer_4qvQ · 2024-12-09
**Job Search Platform**

**Confidence:** 5
**Significance Of Results:** 5
**Overall Quality:** 5

**Compliance With Template:**

5: Very High Quality – The article contains all the required sections, which are written in a very detailed, clear, and error-free manner. The structure is professional and meets expectations, and the content adheres to the highest substantive and formal standards.

**Description Of Results:**

4: High Quality – The results are described in detail and supported by usage examples or evaluations. The description is reliable but may lack full depth of analysis.

**Feedback On Consistency:**

The article provides a coherent narrative that moves logically from the initial problem statement—inefficiencies in recruitment processes—to the technical solutions proposed, such as custom quizzes and microservices architecture. The subsequent presentation of results aligns with these initial objectives, highlighting how implemented features address earlier identified challenges. Although the conclusions would benefit from incorporating more user-centric data and competitive comparisons, the progression from problem analysis to proposed solution, implementation details, and future directions remains consistent and logically structured throughout. In addition some screens or visual explanation of use cases could allow to better understand the idea around the app.

**Potential For Development:**

The article acknowledges areas for future expansion and refinement. Suggestions include developing a native mobile application to increase accessibility, integrating advanced AI for more sophisticated recommendations, and partnering with established professional networks to broaden the platform’s reach. Expanding language support, refining user experience through empirical validation, and addressing unmet user needs indicate that the project’s foundation is robust, leaving ample room for practical enhancements and subsequent iterations. Such forward-looking considerations position the platform for ongoing improvement and potential adoption in real-world recruitment scenarios.

Summary:
This platform stands out for its scalable architecture, quiz-based pre-evaluation, and interactive analytics, making it an interesting alternative to conventional recruitment tools. However, it falls short by not incorporating cutting-edge features that many competitors are exploring, such as AI-driven interviews and more advanced methods for verifying a candidate’s depth of knowledge. As a result, while the app holds promise, it may struggle to position itself as a leader in the rapidly evolving recruitment technology market.

**Project Nature Evaluation:**

The project exhibits strong hallmarks of an engineering effort. Its use of a microservices architecture, reliance on containerization (Docker and Kubernetes), and usage of various programming languages for specialized functions demonstrate a purposeful application of technical methods. Furthermore, the platform aims to solve a practical recruitment challenge with measurable utility—improving hiring efficiency, communication, and data-driven insights. These aspects, combined with scalability considerations and integration of analytics and recommendation algorithms, affirm that the project is firmly rooted in engineering principles and problem-solving methodologies.
I strongly recommend that authors review the mock.pl, which provides quite interesting features in that area and may help them better define future directions.

**Technical Language Precision:**

5: Very High Quality – The language is entirely appropriate for a technical report. All terms are used correctly and precisely, and the style is professional, clear, and coherent, without any errors or ambiguities.

---

### Decision · Program_Chairs · 2024-12-10

Accept (Poster)